# The Current State of Traumatic Brain Injury Biomarker Measurement Methods

**DOI:** 10.3390/bios11090319

**Published:** 2021-09-07

**Authors:** Alyse D. Krausz, Frederick K. Korley, Mark A. Burns

**Affiliations:** 1Biomedical Engineering Department, University of Michigan, Ann Arbor, MI 48109, USA; 2Emergency Medicine, Michigan Medicine, University of Michigan, Ann Arbor, MI 48109, USA; korley@umich.edu; 3Chemical Engineering Department, University of Michigan, Ann Arbor, MI 48109, USA

**Keywords:** traumatic brain injury, biomarkers, microfluidics, electrochemical detection, optical detection, immunosensors

## Abstract

Traumatic brain injury (TBI) is associated with high rates of morbidity and mortality partially due to the limited tools available for diagnosis and classification. Measuring panels of protein biomarkers released into the bloodstream after injury has been proposed to diagnose TBI, inform treatment decisions, and monitor the progression of the injury. Being able to measure these protein biomarkers at the point-of-care would enable assessment of TBIs from the point-of-injury to the patient’s hospital bedside. In this review, we provide a detailed discussion of devices reported in the academic literature and available on the market that have been designed to measure TBI protein biomarkers in various biofluids and contexts. We also assess the challenges associated with TBI biomarker measurement devices and suggest future research directions to encourage translation of these devices to clinical use.

## 1. Introduction

Traumatic brain injury (TBI) is considered to be a “silent epidemic” as public awareness remains low and no new treatments have been approved in the past 30 years [1]. TBI is the leading global cause of morbidity and mortality in children and young adults, mainly resulting from motor vehicle accidents, sports injuries, falls, and blasts [2]. An estimated 69 million (95% confidence interval 64–74 million) people across the globe suffer a TBI annually [3]. In the United States alone, approximately 2.5 million people sustain a TBI each year, and 52,844 of these individuals die of their injuries [4]. Furthermore, 70–90% of the annually reported TBIs (1.75–2.25 million) in the United States are considered to be mild, but even a mild TBI (mTBI) can have a significant effect on an individual’s quality of life [1,5,6]. In fact, 47.4% of individuals aged 40 years or older in the United States with a history of head injury are living with a disability, corresponding to 11.4 million people [7].

The high rates of morbidity and mortality associated with TBI are partially due to limited tools for diagnosis and classification that fail to capture the heterogeneity of TBI-related injuries and make the development of new treatments challenging [1,8]. Current diagnostic methods rely on neurological examination (Glasgow Coma Scale (GCS)) and neuroimaging techniques (computed tomography (CT)) to assess the severity of the injury and the extent of brain tissue damage. The GCS is used to classify TBI into mild (GCS 13–15), moderate (GCS 9–12), or severe (GCS 3–8) by examining an individual’s motor, verbal, and eye-opening response. The GCS was first proposed in 1974 to aid in the clinical assessment of unconsciousness and has since been used extensively in trauma and critical illness classification schemes [9]. However, the GCS assessment can be confounded by factors such as other traumatic injuries, age, medications, and intoxication [10]. Furthermore, classification of TBI into mild, moderate, or severe has limited utility in the prediction of outcomes for an individual patient [11]. Computed tomography (CT) can be used to visualize tissue damage and to assess the need for neurosurgical intervention. However, mTBI is often not associated with abnormalities on head CTs, incurring unnecessary medical costs and exposing patients to unnecessary radiation [10,12].

Given the limitations in the current TBI assessment methods, there is a need for tools that can provide precise diagnoses and prognoses across all categories of TBI and enable accurate and rapid triage and treatment. The measurement of biofluid-based protein biomarkers released from damaged brain cells into systemic circulation following a TBI has been proposed to fulfill this clinical need.

## 2. Traumatic Brain Injury Biofluid Biomarkers

Generally, biomarkers can be classified into three main categories: diagnostic, prognostic, and predictive [13]. A diagnostic marker can be used to detect the presence of a disease or condition of interest such as blood sugar or hemoglobin A1c in Type 2 diabetes mellitus [14]. A prognostic marker is used to determine the likelihood of a clinical event or disease progression. For example, increasing concentrations of prostate-specific antigen can be used to assess the likelihood of cancer progression [15]. A predictive marker can be used to identify individuals who are more likely to experience favorable or unfavorable effects from a therapy. BRCA1/2 mutations are often used to identify patients who are likely to respond to PARP inhibitors [16]. The classification of protein biomarkers for TBI assessment is ongoing, and it is likely that the same biomarker could be classified into multiple categories. For example, glial fibrillary acidic protein (GFAP), a marker of astroglial damage, has been studied as both a diagnostic and prognostic marker for TBI [17,18].

Regardless of classification, a biomarker must be sensitive and specific [1,19]. Clinically, sensitivity refers to a biomarker’s ability to identify patients who have a disease or condition, while specificity refers to the biomarker’s ability to identify patients who do not have the disease [20]. These parameters are influenced by a variety of factors including assay performance, comorbid conditions, and where and in what quantities a biomarker is released within the body. In the case of TBI, biomarkers should be measured that are released primarily from the central nervous system (CNS), and the measurement methods used should be able to accurately quantify protein concentrations on the order of tens of picograms per milliliter. To capture the diversity of injured cell types, panels of biomarkers have been proposed to assess traumatic brain injuries [5,21,22]. As of June 2021, only GFAP and ubiquitin c-terminal hydrolase L1 (UCH-L1) are FDA cleared to aid in determining the need for evaluation with a head CT scan in the United States [23]. S100B is currently used for the same purpose in Europe and may result in a decrease in head CT utilization [24]. Thorough discussions of the current state of TBI biofluid biomarkers can be found in recent reviews by Wang et al. (2018), Gan et al., and Slavoaca et al. [25,26,27].

This review summarizes biomarker measurement methods that target TBI assessment. We categorize these devices both by stage of development (early-stage (academia) or late-stage (commercially available)) and by detection mechanism. There is a large body of literature on protein measurement devices and TBI biomarkers have been studied in the context of other conditions, so we focused on the measurement of proteins in the context of TBI. For example, there are many studies in which IL-6 was measured, but most are related to inflammation and sepsis [28,29,30]. The scope of this review ensures that all included devices were designed to measure biomarker concentrations in a range that is clinically relevant to TBI and that any validation experiments were performed using clinical samples from TBI patients. These parameters allow direct comparisons to be made between devices and methods. Table 1 provides a detailed list of the TBI protein biomarkers measured by the devices discussed in this review.

## 3. Design Considerations for TBI Protein Biomarker Measurement Devices

TBI biofluid biomarkers have the potential to revolutionize the way TBI is assessed and treated beyond evaluating the need for a head CT [82]. Biomarkers could be used by military personnel for rapid triage in the field, by paramedics to assess a patient’s condition prior to arrival at a hospital, by clinicians to monitor a patient’s response to neuroprotective treatments and provide accurate prognoses, and by clinical research teams to accurately enroll and group patients for clinical trials. However, for biomarkers to be useful in each of these settings, a rapid, robust, accurate, and easy-to-use measurement platform is needed. The ideal TBI protein biomarker measurement device would possess the following characteristics:Usability Considerations
○Usable by untrained personnel;○Functional in austere conditions (extreme temperature and humidity);○Requires minimal hands-on time.Assay Considerations
○Requires minimal sample preparation;○Assay performed on a drop (50 µL) of capillary whole blood obtained via a fingerstick;○Simultaneous measurement of at least 2 and up to 10 biomarkers (multiplexing);○Precise readings within the same run (intra-assay coefficient of variation (CV) ≤ 10%) and between runs (inter-assay CV ≤ 15%);○Linear range extends across the concentrations of interest for a specific biomarker;○Lower limit of detection (LLOD) is below the cutoff concentration used to distinguish a physiological concentration of a biomarker from a concentration indicative of TBI;○Results obtained in less than 15 min (timeframe based on current clinical management workflows).Mass Production Considerations
○Reagents stable for a year or longer;○Inexpensive to manufacture.Clinical Utility Considerations
○Portable;○Accurately identifies patients with a TBI;○Accurately identifies patients without a TBI.

Achieving just one of these ideal characteristics is a challenge, so device development often focuses on innovating in one of the areas of consideration listed above. We discuss how the sensors covered in this review have achieved these ideal characteristics and which areas present opportunities for continued innovation in the Discussion and Outlook section.

## 4. Early-Stage Measurement Methods for Detection of TBI Protein Biomarkers

Our review identified 24 sensor designs and measurement methods in the early stages of development that specifically listed TBI as an application. To the best of our knowledge, these sensors have only been described in the academic literature and are not yet commercially available. We have categorized these sensors according to detection method: electrochemical (including field-effect transistors (FET)), optical, surface-enhanced Raman spectroscopy (SERS) (a specific type of optical detection), and surface acoustic wave (SAW). A summary of the assay characteristics for each early-stage sensor can be found in Table 2.

### 4.1. Electrochemical Detection

Electrochemical sensors involve the transduction of chemical changes into an electrical signal often using a three-electrode system (counter, working, and reference electrodes). Typically, a counter electrode applies a stimulating signal to the working electrode on which a chemical reaction takes place. In the case of TBI biomarker measurement systems, the working electrode is functionalized either with enzymes that convert the target biomarker into a product that is electrically detectable or antibodies that bind the biomarker of interest. The signal of the working electrode is measured in relation to the reference electrode using impedimetric, amperometric, and potentiometric techniques.

Electrochemical impedance spectroscopy (EIS) and impedance-time (Z-t) techniques have been used to quantify norepinephrine (NE), GFAP, neuron specific enolase (NSE), S100B, and tumor necrosis factor-α (TNF-α) [44,72,73]. For all measurements, the electrochemical cell consisted of a gold disk working electrode, a platinum wire counter electrode, and an Ag/AgCl wire reference electrode. The surface of the gold disk electrode was functionalized either with an enzyme to detect NE [72,73] or antibodies for either GFAP, NSE, S100B, or TNF-α [44]. EIS was used to quantify the biomarker concentrations by applying an AC potential with changing frequencies to stimulate the solution in the electrochemical cell while measuring current. Impedance and phase were then calculated in real time based on the measured current [73]. The Z-t technique, which is the EIS technique at a static frequency, was also used to quantify the TBI biomarkers. Using Z-t simplifies the electronics and makes this electrochemical system more attractive for point-of-care settings [72]. For the GFAP, NSE, S100B, and TNF-α measurements, the authors listed the detection limits as 2–5 pg/mL in buffer and 14–67 pg/mL in 90% rat whole blood. However, it was unclear whether EIS or Z-t was used to obtain these LLODs and which limits corresponded to which assay [44]. Furthermore, separate electrochemical cells were used to quantify each analyte, but a single cell functionalized with antibodies for all four biomarkers could be used in the future.

The EIS technique was also employed by Arya et al. who examined how electrode geometry and electrode functionalization protocols influenced the label-free detection of GFAP [43]. A gold microdisk electrode array (MDEA) consisting of six, 100 µm diameter microdisks and a gold macroelectrode with a comb structure (MECS) consisting of an electrode that was 5 µm wide with 10 µm of spacing between comb fingers were fabricated on an oxidized silicon wafer [83]. The electrodes were functionalized with anti-GFAP antibodies using two different protocols. The optimized functionalization protocols resulted in a LLOD of 1 pg/mL for both the MDEA and the MECS in spiked buffer solutions, which is below the cutoff for assessing mTBI [43]. However, this limit of detection could increase when the complexity of the sample matrix increases (i.e., serum, plasma, or whole blood) due to non-specific interactions with the antibodies.

An interesting application of electrochemical sensors to the diagnosis of TBI is amperometric enzyme-based logic systems [54,55,61,62,84,85]. These sensors digitally process multiple biomarker signals by applying a constant potential and measuring the corresponding current between the working and counter electrode to produce a final YES/NO response through Boolean logic gates composed of enzymatic reactions [84]. Pita et al. developed a system consisting of an AND and an XOR logic gate to distinguish between hemorrhagic shock (HS), TBI, and an ischemic state [62]. The AND gate took oxygen and NE as inputs and the XOR logic gate took glucose and lactate as inputs. The outputs from each logic gate were measured using a glassy carbon working electrode, platinum wire counter electrode, and Ag/AgCl reference electrode with physiologically normal concentrations of the inputs assigned as logic 0 and pathological concentrations assigned as logic 1. If the AND gate output was 1 and the XOR gate output was +1, signaling that NE and lactate were at pathological concentrations, then a TBI was indicated. Manesh et al. also measured glucose, lactate, and NE to distinguish between HS and TBI [61]. In this case, an AND gate was used that accepted lactate and NE as inputs and an IDENTITY gate was used that accepted glucose as an input. This system also indicated a TBI if NE and lactate were at pathological concentrations such that the AND gate returned a logical 1 and the IDENTITY gate returned a logical 0. Halámek et al. streamlined the logic such that the output of a single gate could be used to determine the presence of a TBI [54]. An AND gate was used that took glutamate and NE as inputs. The output from the logic gate was measured using a screen-printed three electrode system where the working and counter electrode were made using carbon-based ink and the reference electrode was printed using Ag/AgCl-based ink. A TBI was indicated if the AND gate returned a logic value of 1, signaling that both NE and glutamate were present at pathological concentrations. Finally, Zhou et al. demonstrated the operation of enzyme logic gates with human serum samples [55]. A NAND gate was used that had glutamate and lactate dehydrogenase as inputs. The output from the NAND gate was measured using the same electrode employed by Pita et al. [62]. If the NAND gate output a 0, indicating that glutamate and lactate dehydrogenase were at pathological concentrations, then a TBI was implicated. Overall, these enzyme-logic gate sensors are a unique application of electrochemical detection and have been shown to function in both buffer and serum solutions. However, they are inherently semi-quantitative and cannot be applied to all protein biomarkers. For a further discussion of enzyme-based logic systems, the authors refer the reader to reviews by Wang and Katz [84,85].

Amperometric techniques can also be used to achieve continuous monitoring of TBI biomarkers. Gunawardhana and Lunte developed a PDMS sensor that uses microdialysis and microchip electrophoresis with end-channel amperometric detection to continuously measure adenosine and its metabolites (inosine, hypoxanthine, and guanosine) [33]. The PDMS microchip was reversibly sealed such that a 33 µm diameter carbon fiber microelectrode could be easily placed and reused. Furthermore, 1 mm diameter Pt and Ag/AgCl wires were used as the counter and reference electrodes, respectively. The 5 cm long separation channel in the microchip achieved separation of all four purines in ~85 s. Future work will focus on lowering the limits of detection for each analyte such that the device could be used to continuously monitor the concentration of adenosine and its metabolites in brain extracellular fluid after severe TBI. However, adenosine has not been extensively studied in the context of TBI, so the clinical utility of this biomarker remains to be determined.

Khetani et al. used differential pulse voltammetry (DPV), a subclass of amperometry, to make the testing of TBI biomarkers portable. DPV involves applying potential pulses with a linear ramp in potential and measuring the corresponding current as a function of potential. The authors developed µDrop, a low-cost potentiostat coupled with functionalized electrodes (immuno-biosensors) for the measurement of cleaved tau (C-tau) and neurofilament light (NF-L) [36]. The µDrop consists of eight DPV modules, allowing the measurement of eight immuno-biosensors in parallel (Figure 1). The immuno-biosensors in the µDrop system were formed by functionalizing the surface of sputtered gold electrodes with antibodies for C-tau and NF-L. The reference electrode was Pt, and the substrate for the electrodes was glass. The µDrop system was used to measure six serum samples from patients with a confirmed TBI and outperformed an ELISA in specificity and sensitivity. Wang et al. also used DPV to quantify GFAP with screen-printed carbon electrodes (SPCEs) coated with a molecularly imprinted polymer that contained multiwalled carbon nanotubes (MWCNTs) [50]. However, this system was not portable and the LLOD of 0.04 µg/mL was not clinically relevant to the diagnosis or monitoring of TBI.

FET-based biosensors are a specific type of potentiometric electrochemical biosensor. These sensors consist of a source and a drain connected by a gate. Typically, antibodies for an analyte of interest are conjugated on the gate, and the gate voltage is affected by the concentration of the analyte bound to the antibodies [86]. Song et al. developed an organic field effect transistor (OFET) for measuring GFAP [49]. The design featured an extended solution gate achieved by incorporating polyethylene glycol (PEG) into the antibody layer to extend the Debye screening length (distance beyond which voltage changes are not sensed) (Figure 2). The extended gate allows the sensing area to be separated from the organic semiconductor and to work without a reference electrode, simplifying and stabilizing the sensor. However, the LLOD of the sensor was only 1 ng/mL, above the cutoff for a mTBI (GCS 13–15). Huang et al. designed an organic thin film transistor (a special type of metal-oxide semiconductor field-effect transistor (MOSFET)) for GFAP consisting of both p-channel and n-channel semiconductors to enable identification of electrical crosstalk and false-positives [45]. The LLOD was the same as the device from Huang et al. at 1 ng/mL.

For a discussion of electrochemical sensing of TBI biomarkers beyond that presented here, the authors refer the reader to the recent review by Pankratova et al. [87].

### 4.2. Optical Detection

Sensing techniques such as fluorescence, chemiluminescence, and colorimetry relate the intensity of light or color generation to the concentration of an analyte. Optical detection techniques are often used as the readout for sandwich immunoassays, a well-established method for quantifying protein biomarkers. The optical sensing methods developed to measure TBI biomarkers involve binding reactions that occur in a centrifuge tube, reactions that occur on chip, and reactions that are multiplexed.

Kim and Searson developed an immunoassay for S100B using magnetic beads as the substrate for the capture antibody and a quantum dot as the fluorescent readout [77]. By cleaving the quantum dots from the sandwich complex on the magnetic beads, they achieved a LLOD of 10 pg/mL in human serum. A similar method using carbon dots was developed by Ma et al. for detecting GFAP [47]. Han et al. used a ratiometric fluorescent readout to quantify S100B by binding peptide functionalized nanoprobes (carbon dots and gold nanoclusters) to the analyte [76]. When S100B bound to the peptide on the nanoprobe, the fluorescence of the gold nanoclusters was quenched while the fluorescence from the carbon dots remained unchanged. The intensity ratio of the carbon dots to the gold nanoclusters correlated with the concentration of S100B in the sample.

The fluorescent immunoassay methods described in Kim and Searson, Ma et al., and Han et al. were carried out in a centrifuge tube. Immunoassays must be coupled with an analysis and readout device to be useful in point-of-care settings. To that end, Bradley-Whitman et al. developed a lateral flow strip for visinin-like protein 1 (VILIP-1) with a colorimetric readout that they tested using serum samples from Sprague Dawley rats [81]. Apori and Herr used a microfluidic device for immunosubtraction to simultaneously detect S100B and C-reactive protein (CRP) in cerebrospinal fluid (CSF) [40]. A polyacrylamide gel electrophoresis (PAGE) separation channel was incorporated into a microfluidic device capable of sample enrichment, fluorescence labeling, and mixing of the sample and capture antibody. The PAGE channel incorporated a step-decrease in separation matrix pore-size at the start of the channel that excluded the immune-complex formed by the target protein and capture antibody. If the capture antibodies were absent, then the PAGE assay contained all protein peaks, but if the immune-complex was present, the assay contained no protein peaks, hence immunosubtraction. The fluorescent readout of the PAGE assay came from labeling the target proteins with Quant-iT dye that undergoes fluorescent enhancement when bound to proteins.

Multiplexing in TBI assessment is important as the simultaneous assessment of two or more biomarkers increases the sensitivity and specificity of the assay. Krausz et al. have recently developed a variable height device capable of passively multiplexing bead-based immunoassays for GFAP, interleukin-6 (IL-6), and interleukin-8 (IL-8) [46]. The variable height device gradually decreases in height between the channel inlet and outlet, causing assay beads to become physically trapped where their diameter matches the channel height. By using a different diameter bead for each assay, distinct bands form within the channel, similar in appearance to an electrophoretic separation. A different colored quantum dot was used as the fluorescent label for each assay to visualize potential bead crossover between detection bands, since smaller beads flow through bands of larger beads as they are forming (Figure 3). The observed bead crossover was minimal, allowing a single quantum dot to be used for all three assays in the future, which would simplify the development of portable optics. The variable height platform is highly flexible as additional assays can be multiplexed simply by adding in beads of a different diameter, allowing the platform to keep pace with advances in TBI biomarker discovery and validation.

### 4.3. Surface-Enhanced Raman Spectroscopy (SERS)

SERS is a specialized optical technique that utilizes the phenomenon in which inelastic light scattering by molecules is enhanced (by factors of 10^8^ or more) when the molecules are adsorbed to metal surfaces, such as gold nanoparticles [88]. This enhancement of light scattering can offer single-molecule resolution as well as molecular fingerprints for label-free identification of analytes of interest [89]. TBI biomarker measurement devices use a variety of SERS probe designs to enhance inelastic light scattering.

Rickard et al. developed an array of gold-coated nanopillars (500 nm tall) that they incorporated into a PDMS microfluidic device for the multiplexed analysis of N-acetylasparate (NAA), S100B, and GFAP (Figure 4) [48]. A fingerprick sample of whole blood was loaded into a microfluidic device that separated the red blood cells from the plasma, which then flowed across the SERS substrate (gold-coated nanopillars). The SERS spectrum was acquired using a miniaturized optics system and analyzed for pre-established biomarker fingerprints. The system was used to analyze blood samples collected from people with TBI and healthy volunteers and to temporally profile NAA concentrations post-TBI. This SERS-based system offers rapid and label-free detection. However, any imperfection on the SERS substrate significantly degrades analytic performance, which may limit the system’s deployment in point-of-care settings [90].

Gao et al. developed a lateral-flow assay for neuron-specific enolase (NSE) with SERS-based detection [68]. The system functioned as a typical lateral flow assay except that the detection antibodies were conjugated to Au nanostar@Raman Reporter@silica sandwich nanoparticles consisting of a Raman reporter sandwiched between an Au nanostar and a thin silica shell. When used to analyze clinical plasma samples from TBI patients, the SERS lateral-flow assay compared favorably to an ELISA with no significant difference between the two methods. Gao et al. later achieved a lower limit of detection when measuring S100B by conjugating the lateral-flow assay capture antibodies to a gold nano-pyramid array on a quartz substrate [75]. When the detection antibody/S100B/capture antibody sandwich was formed, SERS probes on the detection antibodies were brought into proximity to the gold nano-pyramids, creating “hot spots” that amplified the SERS signal and increased the sensitivity of the assay.

Wang et al. used a similar method to quantify NSE and S100B [70]. Hollow gold nanospheres were functionalized with detection antibodies either for NSE or S100B, and a glass slide with layers of hollow gold nanospheres was functionalized with capture antibodies. When the hollow gold nanospheres were brought into proximity by binding with the analyte of interest, the SERS signal was enhanced. There was found to be no significant difference between the gold nanosphere assay and an ELISA when the comparison was performed using clinical serum samples. Li et al. similarly used the proximity of antibody functionalized Au nanocages contained within a lateral flow glass-hemostix to measure NSE in diluted plasma samples (80% plasma and 20% PBS solution) [69]. The Au nanocage method also compared favorably to an ELISA.

### 4.4. Surface Acoustic Wave (SAW)

SAW biosensors detect frequency changes that occur due to mass-loading effects in an acoustic wave traveling along a piezoelectric crystal surface [91]. Agostini et al. developed an ultra-high-frequency surface-acoustic-wave (UHF-SAW) lab-on-a-chip to measure GFAP in a bovine serum albumin matrix [42]. The UHF-SAW device consisted of a SAW-resonator and waveguide adhered to PDMS microchannels for liquid manipulation (Figure 5) [92]. The SAW-resonator was functionalized with antibodies for GFAP, and the shifts in resonance frequency depended on the concentration of GFAP in the sample. The goal of this work was to determine the optimal functionalization protocol, so GFAP was not measured at clinically relevant concentrations. However, the UHF-SAW device is equipped with four resonators, allowing for multiplexed measurements in the future.

## 5. Late-Stage Measurement Methods for Detection of TBI Protein Biomarkers

There are currently three sensor systems for TBI protein biomarkers that can be considered as late-stage, meaning that these devices are either being used in clinical studies or are already on the market for measuring biomarkers in clinical practice. Two of these devices use optical detection and one uses electrochemical detection to quantify biofluid biomarkers. A summary of the assay characteristics for each late-stage device can be found in Table 3.

### 5.1. Banyan BTI^TM^

The Banyan Brain Trauma Indicator (BTI) from Banyan Biomarkers is a traditional test kit for GFAP and UCH-L1 consisting of 96-well plates coated with capture antibodies for each biomarker. A trained technician must manually perform the assay and insert the plates into a reader to quantify the chemiluminescence and determine the biomarker concentrations [93]. The kit was FDA cleared in February 2018 via the de novo pathway and is intended to aid in the evaluation of patients 18 years of age or older with a suspected GCS 13–15 TBI (mTBI) [115]. A positive result occurs if one or both of the GFAP and UCH-L1 concentrations are above the cutoff values (22 pg/mL for GFAP and 327 pg/mL for UCH-L1) and necessitates a CT scan. A negative result is associated with the absence of lesions on a CT scan, ruling out the need for a scan. However, the Banyan BTI^TM^ test takes over 2 h to run, potentially rendering the results unactionable within current clinical management workflows. The BTI^TM^ has been useful in several clinical studies to establish the kinetics [95], diagnostic accuracy [99], and association with CT findings [96,100,104,105] of GFAP and UCH-L1, but the treatment decisions did not depend on the test results. To improve the assay time and render the test usable for making treatment decisions in real time, Banyan Biomarkers provided a non-exclusive license of their TBI biomarkers to Abbott [116].

### 5.2. Abbott i-STAT Alinity

The TBI Plasma cartridge for the Abbott i-STAT Alinity was FDA cleared via the 510(k) pathway in January 2021 using the Banyan BTI^TM^ as the predicate device [117]. The cartridge measures GFAP and UCH-L1 amperometrically using gold working electrodes and a Ag/AgCl reference electrode fabricated on a silicon substrate [118]. The i-STAT Alinity is portable and is designed to be used with the TBI Plasma cartridge in clinical laboratory settings by trained personnel [106]. The i-STAT provides results 15 min after loading a plasma sample onto the cartridge, allowing for the test to be used clinically to assist in determining the need for a head CT scan. Similarly to the Banyan BTI^TM^, a positive result occurs if one or both of the GFAP and UCH-L1 concentrations are above the cutoff values (30 pg/mL for GFAP and 360 pg/mL for UCH-L1) [106]. A positive result necessitates a head CT scan while a negative result rules out the need for a head CT [117]. TRACK-TBI (Transforming Research and Clinical Knowledge in Traumatic Brain Injury) studies using the i-STAT have recently concluded [17,23], and the device is set to be deployed in select hospitals in the United States by the end of 2021.

As of now the i-STAT Alinity TBI cartridge requires a plasma sample, but Abbott is working on a test that utilizes a whole blood sample, which would allow the device to be used at the point-of-care in healthcare settings [119]. If GFAP, UCH-L1, or other TBI biomarkers are eventually used for more than ruling out the need for a head CT scan, the Abbott i-STAT Alinity could potentially be adapted for use in field settings, since the device is already portable.

### 5.3. Quanterix Simoa^®^

The Quanterix Simoa^®^ bead technology has been used in several clinical studies of TBI protein biomarkers [114]. This technology enables the measurement of single protein molecules by confining assay beads with completed sandwich immunocomplexes in wells of extremely small volume (~50 fL) (contained on the Simoa^®^ disc) [120]. The immunocomplexes are a conventional ELISA (enzyme-linked immunosorbent assay), but the wells confine the fluorophores generated by the enzyme, leading to a high local concentration of fluorescent signal. At extremely low protein concentrations, the ratio of protein to beads is less than 1:1, so if protein is present, the beads are labeled with a single enzyme. A digital immunoassay can be performed by calculating the ratio of wells that have a fluorescent signal to those that only contain a bead and then correlating this ratio to the protein concentration in the sample [120]. These assays can also be multiplexed by labeling microbeads with their own fluorescent signatures. The bead fluorescence and the enzyme-generated fluorescence are both measured to determine the signal for each biomarker in the multiplex [121]. Quanterix developed laboratory analyzers, such as the HD-1 and HD-X, to fully automate these assays and provide users with sample-in, results-out workflows [122].

Quanterix has developed assays for many different TBI biomarkers, including GFAP, UCH-L1, tau, NF-L, and NSE [107]. However, these assays are for research use only, and the results cannot be used to make clinical decisions. Quanterix is currently enabling clinical research into different contexts of use for emerging TBI biomarkers [111,112,113], but an FDA cleared assay will need to be available before clinical translation can occur.

## 6. Discussion and Outlook

We have provided a detailed overview of sensors developed for detection of traumatic brain injury biofluid protein biomarkers with a focus on those described in the academic literature as well as commercially available devices. These sensors used electrochemical, optical, and acoustic techniques to quantify TBI biomarkers. Electrochemical sensors have been successfully used as monitoring devices for blood glucose, so there is a precedent for deploying these sensors at the point-of-care and in the field [123]. With optimized limits of detection, multiplexing, and portable electronics, electrochemical biosensors could potentially be used by patients to determine whether to seek treatment for a head injury and to monitor their recovery after seeking medical treatment. Sandwich immunoassays with an optical readout have been widely used to quantify protein biomarkers in a laboratory setting with low limits of detection and high sensitivity and specificity. The challenge for deploying these methods at the point-of-care is designing optics that are sufficiently small and robust and incorporating reagent mixing and washing steps onto a chip. SERS, a specialized optical technique, offers label-free detection with single-molecule resolution, but the fragility of the substrates and complexity of the optics may make this detection method difficult to realize at the point-of-care. SAW biosensors are attractive for TBI assessment since they offer label-free detection, but the materials and fabrication can be expensive. Furthermore, any changes to the path of the wave will result in a change in output, potentially limiting the deployment of these sensors in field settings.

Several of these sensors were designed for use at the point-of-care, but further development is needed to move these devices out of a healthcare setting and into the field. A device that can be used both inside and outside of a hospital would enable rapid triage at the site of an injury, monitoring and assessment in an ambulance, and serial measurements in a hospital setting to monitor a patient’s response to neuroprotective treatments and to predict outcomes.

To achieve this ideal use case, development of biomarker measurement devices and clinical validation of protein biomarkers must occur simultaneously. Biomarkers must be cleared for determining whether to seek medical treatment for an injury, to enable rapid triage at the site of an injury; for determining injury severity, to enable monitoring and assessment in an ambulance; and for monitoring patient response to treatment and determining prognoses, to enable the utility of serial measurements in a hospital setting. FDA authorizations will be challenging to obtain if point-of-care devices do not exist to measure TBI protein biomarkers in each of these contexts.

Ideal characteristics for such point-of-care devices are described in Section 3. Most of the devices covered in this review have achieved a relatively short analysis time, low sample and reagent consumption due to the use of microfluidics, and inexpensive manufacturability as microfabrication approaches can be easily scaled. However, clinical parameters in addition to the features of the device itself need to be considered when developing a biomarker measurement device.

The first clinical parameter is the choice of biomarker. GFAP and UCH-L1 are FDA cleared for assessing TBIs in the United States, and S100B is approved in Europe [23,24], but only 9 out of 25 devices described in the literature measured GFAP, 7 devices measured S100B, and not a single device measured UCH-L1. To improve the chances of device translation, the measured biomarkers should already have proven clinical utility. If the purpose of developing a new device is to study a novel, promising biomarker, then it may be advantageous to choose a clinically established detection method, such as electrochemical or optical detection, so that the results of the novel device can be more readily compared to clinical tests. Multiplexed systems would also allow for the exploration of novel biomarkers while simultaneously quantifying proteins with proven utility in TBI evaluation.

After choosing a biomarker to measure, the second clinical parameter to consider is the biofluid used as the sample [124]. Clinically, protein biomarkers have been measured in CSF, serum, and plasma. Abbott is working on a test to measure biomarkers in whole blood, and Krausz et al. demonstrated proof-of-concept GFAP measurements using spiked whole blood from a single donor [46,119]. Rickard et al. demonstrated NAA measurements using fingerstick blood samples, but to the best of our knowledge, no one in industry has measured TBI protein biomarkers in a fingerstick blood sample, even though this sample is ideal for point-of-care biomarker measurements [48]. This may be due to potential analytical interference from hemolysis caused by excessive squeezing or massaging of the puncture site during blood collection [125]. In general, the choice of biofluid dictates the sample collection technique as well as the bioavailability [126] and kinetics [127,128] of the biomarker which in turn determine the sample processing [129] that needs to occur on the measurement device, the necessary limit of detection of the assay, and the time frame in which a device can be used to obtain meaningful protein concentrations. For example, GFAP is present in serum samples at detectable concentrations (>0.030 ng/mL) within 1 h of injury [99]. Therefore, a device to measure GFAP must be used 1 h or more after an injury, requires a quantification limit of 0.030 ng/mL, and requires the ability to separate serum from whole blood. It is worth noting that GFAP might be present in serum sooner than 1 h after the injury, but in the study by Papa et al., samples were collected once a patient arrived at a hospital [99]. A device that is capable of quantifying GFAP in field settings could help to determine the first time point at which the biomarker is present in serum samples.

The choice of biofluid for the assay also determines the third clinical parameter, the context of use, which in turn dictates the usability and robustness requirements of the measurement device. CSF can only be collected in a hospital setting, whereas fingerstick whole blood can be easily collected in the field. Serum and plasma could be used in either a healthcare or field setting depending on the sample preparation capabilities of the biomarker measurement device. Assuming that fingerstick whole blood can be used as the sample for the biomarker measurement and that the device is going to be used in the field by untrained personnel, then the user should only have to perform 1–3 steps to run the test, and the device should withstand falls as well as extreme temperatures and humidity.

To fully assess the interconnectedness of clinical, assay, and device parameters, partnering with clinicians is paramount. Clinicians can determine which biomarkers to measure and what the cutoff concentrations should be, how the device will fit into existing clinical workflows, the required sensitivity and specificity for the test, and the context in which the device will be used clinically (e.g., in the field, in an ambulance, in an emergency department, etc.). These parameters then determine the features of the sensing system such as the detection mechanism (e.g., optical, electrochemical, acoustic, etc.), required robustness, usability, and limit of detection. Once a device exists to monitor biomarkers in the desired context of use, results from clinical validation studies can further refine the device features, allowing the feedback cycle between clinicians and engineers to repeat (Figure 6). Initial clinical validation can take place with a prototype device that does not achieve all the requirements for an ideal device. To accelerate translation, it is important to enter the feedback loop between clinicians and engineers as quickly as possible so that the prototype device can be refined based on clinical feedback.

To achieve testable prototypes as quickly as possible, TBI biomarker measurement device development should focus on multiplexing, assay evaluation with complex samples (serum, plasma, and/or whole blood), usability, and robustness. GFAP and UCH-L1 must be measured together in the United States to assess the need for a head CT scan [23], and panels of three or more biomarkers have been proposed [130,131]. Therefore, a TBI biomarker device is only clinically useful if it can measure at least two biomarkers simultaneously. Most of the devices covered in this review only evaluated spiked buffer solutions. While this approach is useful for optimizing assay parameters, buffer solutions are not reflective of the complexity inherent in biofluids. Any interfering substances present in the complex biofluid matrix must be assessed before the measurement device can be validated clinically. Additionally, a device must be initially usable by trained personnel in a hospital laboratory setting at the minimum. As such, users besides the developers must be able to obtain repeatable biomarker concentrations using the device, and the measurements cannot be influenced by ambient temperature, humidity, or particulates.

A promising step towards incorporating the measurement of biofluid biomarkers for TBI assessment into clinical practice came with the recent FDA clearance of the i-STAT Alinity from Abbott. The device is portable, so it has the potential to be used in multiple contexts beyond the hospital laboratory setting it is approved to be used in as of June 2021. Data obtained using the i-STAT Alinity will inform clinical cutoffs for GFAP and UCH-L1, which can be used to continue to innovate in the development of biomarker measurement tools. With ongoing innovation in sample preparation, usability, and robustness in both academia and industry, biomarkers could be measured at every stage of TBI assessment from the site of the injury to the hospital bedside, enabling precision diagnosis and monitoring and ultimately leading to improved patient outcomes.

## Figures and Tables

**Figure 1 biosensors-11-00319-f001:**
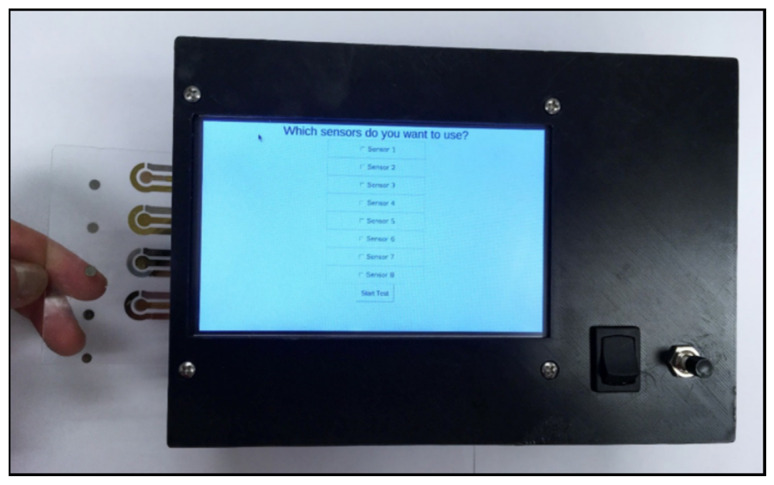
The µDrop system from Khetani et al. designed to perform differential pulse voltammetry to quantify C-tau and NF-L. Reprinted with permission from [36]. Elsevier 2021.

**Figure 2 biosensors-11-00319-f002:**
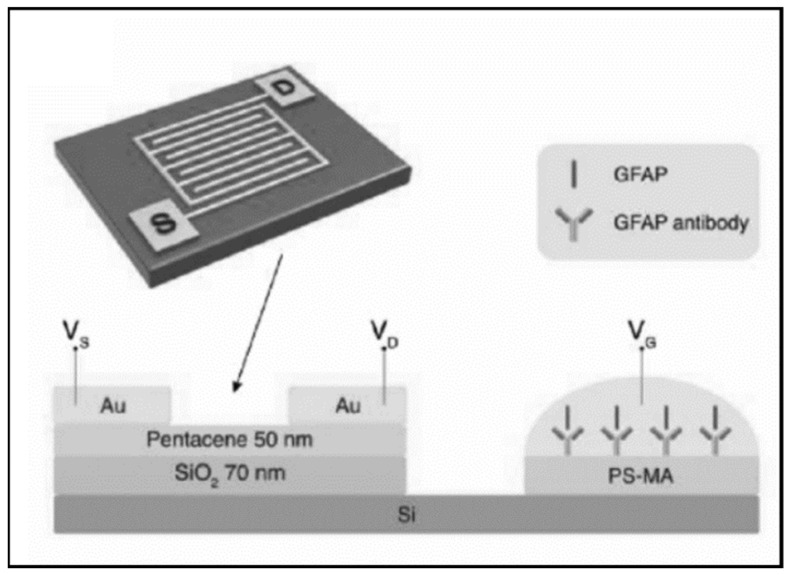
An organic field effect transistor (OFET) with an extended solution gate from Song et al. designed to measure GFAP. Reprinted with permission from [49]. John Wiley and Sons, 2017.

**Figure 3 biosensors-11-00319-f003:**
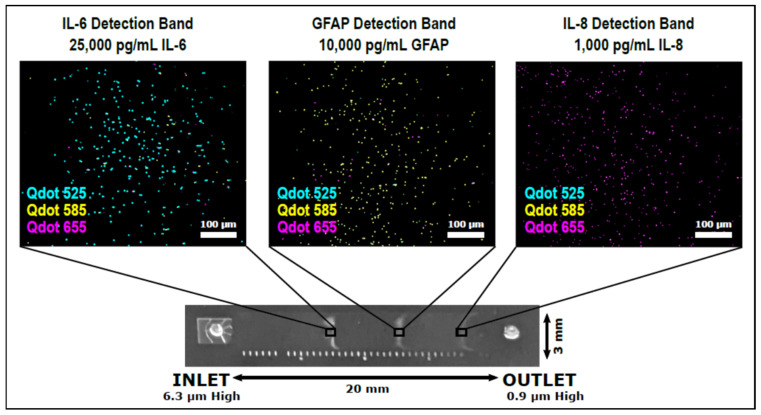
The variable height device from Krausz et al. designed to passively multiplex bead-based QLISAs (quantum dot-linked immunosorbent assays) for GFAP, IL-6, and IL-8 [46].

**Figure 4 biosensors-11-00319-f004:**
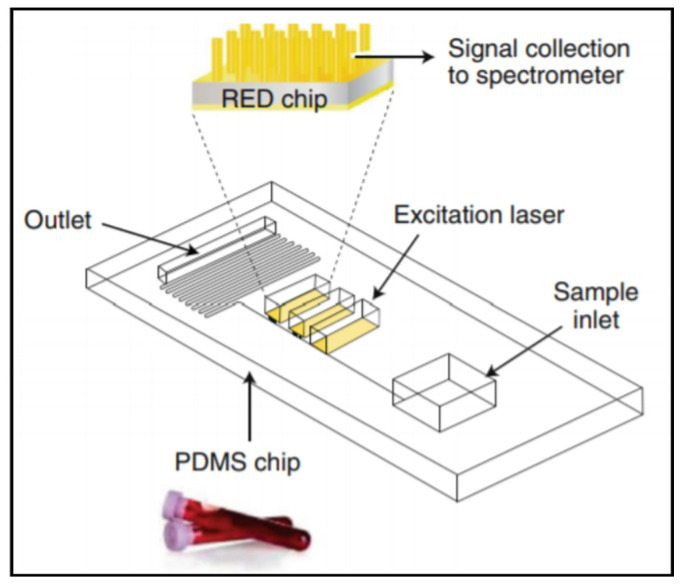
A SERS-based microfluidic device capable of a sample-in, results-out workflow from Rickard et al. Reprinted with permission from [48]. Springer Nature, 2020.

**Figure 5 biosensors-11-00319-f005:**
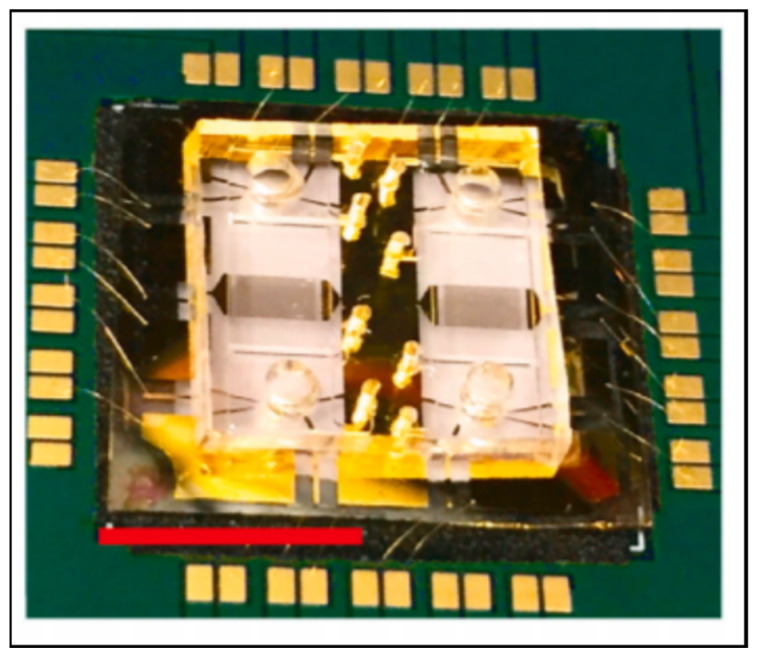
An ultra-high-frequency surface-acoustic-wave chip from Agostini et al. with the ability to multiplex up to four biomarkers. The red scale bar is 1 cm long. Reprinted with permission from [42]. Elsevier, 2021.

**Figure 6 biosensors-11-00319-f006:**
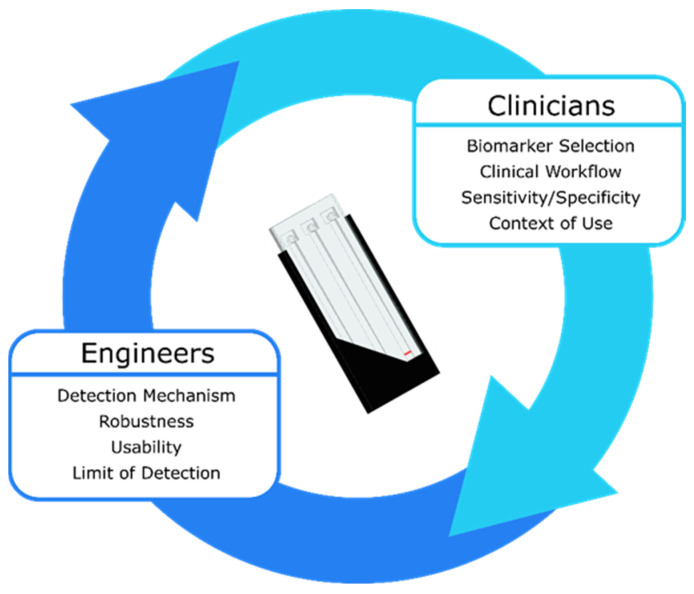
A traumatic brain injury biomarker measurement device can only be developed with constant feedback between clinicians and engineers. The clinical parameters inform the device parameters and vice versa until the optimal device is realized.

**Table 1 biosensors-11-00319-t001:** Traumatic brain injury (TBI) protein biomarkers measured by the devices included in this review.

Biomarker			Physiological Concentration		Relevant Devices
Name	Abbreviation	Injury Information	Normal	Traumatic Brain Injury	
Adenosine	Ado	Severity [31]	4–8 nM in CSF [32]	Severe TBI: 8–16 nM up to 100–800 nM in CSF [31]	Gunawardhana and Lunte [33]
Cleaved tau	C-tau	Neuronal damage (axons) [34]	2.48–66.54 pg/mL in serum [35]	Severe TBI: 36.44–192.34 pg/mL in serum [35]	Khetani et al. [36]
C-reactive protein	CRP	Prognosis [37,38]	0.642–2.785 mg/L in serum [39]	mTBI: 2.110–30.932 mg/L in serum [39]	Apori and Herr [40]
Glial Fibrillary Acidic Protein	GFAP	Astrocyte damage [41]	7–20 pg/mL in plasma [17]	mTBI: 69–1196 pg/mL in plasma [17]	Agostini et al. [42] Arya et al. [43] Cardinell et al. [44] Huang et al. [45] Krausz et al. [46] Ma et al. [47] Rickard et al. [48] Song et al. [49] Wang et al. [50]
Glutamate	-	Neuronal damage (synapses) [51]	0.3–2 µmol/L in brain extracellular fluid [52]	Severe TBI: >20 µmol/L in brain extracellular fluid [53]	Halámek et al. [54] Zhou et al. [55]
Interleukin-6	IL-6	Inflammation [56]	≤1.8 pg/mL in serum [57]	Severe TBI: 0–1100 pg/mL in serum [57]	Krausz et al. [46]
Interleukin-8	IL-8	Inflammation [56]	≤14.6 pg/mL in serum [58]	Severe TBI: 0–2400 pg/mL in serum [57]	Krausz et al. [46]
Lactate	-	Prognosis [59]	6.7–13.9 mg/dL in whole blood [60]	Moderate to Severe TBI: 5.54–11.34 mg/dL in whole blood [60]	Manesh et al. [61] Pita et al. [62]
Lactate dehydrogenase	LDH	Severity [63]	77.3–126.3 IU/L in serum [64]	mTBI: 152.24–247.58 IU/L in serum [64]	Zhou et al. [55]
N-acetylasparate	NAA	Neuronal damage [65]	15.3–36.7 μmol/L in brain extracellular fluid [65]	Severe TBI: 8.8–19.1 μmol/L in brain extracellular fluid [65]	Rickard et al. [48]
Neurofilament light	NF-L	Axonal white matter damage [66]	11–17 pg/mL in serum [66]	Severe TBI: 89–413 pg/mL in serum [66]	Khetani et al. [36]
Neuron Specific Enolase	NSE	Neuronal damage [67]	≤0.15 µg/L in serum [67]	>0.15 µg/L in serum [67]	Cardinell et al. [44] Gao et al. [68] Li et al. [69] Wang et al. [70]
Norepinephrine	NE	Blood-brain barrier (BBB) disruption [71]	185–275 pg/mL in plasma [71]	Severe TBI: >275 pg/mL in plasma [71]	Cardinell and La Belle [72] Halámek et al. [54] Haselwood and La Belle [73] Manesh et al. [61] Pita et al. [62]
S100 Calcium Binding Protein B	S100B	Astrocyte damage [74]	0.06–0.13 µg/L in serum [17]	mTBI: 0.07–0.24 µg/L in serum [17]	Apori and Herr [40] Cardinell et al. [44] Gao et al. [75] Han et al. [76] Kim and Searson [77] Rickard et al. [48] Wang et al. [70]
Tumor Necrosis Factor α	TNF-α	Ischemia [78]	≤4.4 pg/mL in serum [79]	Severe TBI: 0–157 pg/mL in serum [57]	Cardinell et al. [44]
Visinin-like protein 1	VILIP-1	Neuronal damage [80]	21.7–195.3 pg/mL in serum ^a^ [80]	mTBI: 39.3–160.2 pg/mL in serum ^a^ [80]	Bradley-Whitman et al. [81]

Note: ^a^ The reference ranges for visinin-like protein 1 overlap.

**Table 2 biosensors-11-00319-t002:** Summary of early-stage sensors for traumatic brain injury protein biomarker measurement.

Detection Technique	Biomarker(s)	Multiplex	Sample Type	Sample Volume	Analysis Time	Lower Limit of Detection (LLOD)	Range	Ref(s).
Electrochemical (EIS and Z-t)	NE	No	Buffer and 10% rabbit whole blood	50 µL	~90 s	EIS: 98 pg/mL Z-t: 8 pg/mL	1–10,000 pg/mL	[72,73]
Electrochemical (EIS and Z-t)	GFAP NSE S100B TNF-α	No	Buffer and 5%, 12.5%, and 90% rat whole blood and plasma	100 µL	~33 s	Buffer: 2–5 pg/mL 90% whole blood: 14–67 pg/mL ^a^	GFAP: 0.1–2800 pg/mL NSE: 1–25,000 pg/mL S100B: 1–10,000 TNF-α: 0.1–75 pg/mL	[44]
Electrochemical (EIS)	GFAP	No	Buffer	15 µL or 60 µL	~30 min	1 pg/mL	1 pg/mL–100 ng/mL	[43]
Electrochemical (Amperometric)	NE Lactate Glucose (AND and XOR logic gates)	Yes	Buffer	1 mL	~15 min	Glucose: 4 mM Lactate: 2 mM NE: 2.2 nM ^b^	Glucose: 4–30 mM Lactate: 2–13 mM NE: 2.2 nM–3.5 µM ^b^	[62]
Electrochemical (Amperometric)	NE Lactate Glucose (AND and IDENTITY logic gates)	Yes	Buffer	1 mL	~15 min	Glucose: 4 mM Lactate: 2 mM NE: 2.2 nM ^b^	Glucose: 4–30 mM Lactate: 2–13 mM NE: 2.2 nM–3.5 µM ^b^	[61]
Electrochemical (Chronoamperometric)	NE Glutamate	Yes	Buffer	1 mL or 500 µL	~5 min	Glutamate: 40 µM NE: 2.2 nM ^b^	Glutamate: 40–140 µM NE: 2.2 nM–3.5 µM ^b^	[54]
Electrochemical (Chronoamperometric)	Glutamate LDH	Yes	Buffer and human serum	27 µL	~15 s	Glutamate: 40 µM LDH: 0.15 U/mL ^b^	Glutamate: 40–140 µM LDH: 0.15–1 U/mL ^b^	[55]
Electrochemical (Amperometric)	Adenosine Hypoxanthine Guanosine Inosine	Yes	Buffer	~1.5 µL ^c^	~85 s	Adenosine: 25 µM Hypoxanthine: 10 µM Guanosine: 25 µM Inosine: 33 µM	Adenosine: 75–400 µM Hypoxanthine: 20–100 µM Guanosine: 75–400 µM Inosine: 75–150 µM	[33]
Electrochemical (Amperometric)	C-tau NF-L	Yes ^d^	Buffer and human serum	-	~30 min	C-tau (buffer): 0.14 pg/mL C-tau (serum): 0.1 pg/mL NFL (buffer): 0.16 pg/mL NFL (serum): 0.11 pg/mL	Buffer: 1 pg/mL–1 µg/mL Serum: 10 pg/mL–100 ng/mL	[36]
Electrochemical (Amperometric)	GFAP	No	Buffer	-	-	0.04 µg/mL	0.2–10 µg/mL	[50]
SERS	NAA S100B GFAP	Yes ^e^	Human plasma	~50–100 µL ^f^	~2–3 min	NAA: 0.021 pg/mL (0.12 pM) S100B: 3.99 pg/mL (0.19 pM) GFAP: 3.35 pg/mL (0.02 pM)	1 fM–100 nM	[48]
SERS	NSE	No	80% human plasma and 20% PBS	100 µL	~30 min	0.86 ng/mL	1–75 ng/mL	[68]
SERS	S100B	No	80% human plasma and 20% PBS	100 µL	~30 min	5.0 pg/mL	0.1–100 ng/mL	[75]
SERS	NSE	No	80% human plasma and 20% PBS	-	~30 min	0.36 ng/mL	0.5–85 ng/mL	[69]
SERS	NSE S100B	Yes	Human serum	-	-	NSE: 0.1 ng/mL S100B: 0.06 ng/mL	0.2–22 ng/mL	[70]
SAW	GFAP	No	Buffer and bovine serum albumin	200 µL	-	35 pM (in bovine serum albumin) ^g^	-	[42]
Electrochemical (FET)	GFAP	No	Buffer	100 µL	~30 min	1 ng/mL	0.8–400 ng/mL	[45]
Electrochemical (FET)	GFAP	No	Buffer	-	~30 min	1 ng/mL	0.5–100 ng/mL	[49]
Optical Detection (Fluorescence)	S100B CRP	Yes	Ovalbumin and CSF	5 µL	~5 min	S100B (CSF): 65 nM CRP (CSF): 3.25 nM	S100B (ovalbumin): 30 pM–1 µM ^h^	[40]
Optical Detection (Colorimetric)	VILIP-1	No	Rat serum	10 µL	~20 min	5.5 pg/mL	~2–50 pg/mL ^i^	[81]
Optical Detection (Fluorescence)	S100B	No	Human serum	100 µL	~1 h	10 pg/mL	0.1–3 ng/mL	[77]
Optical Detection (Fluorescence)	GFAP	No	Buffer and human serum	200 µL	~90 min	25 pg/mL (buffer)	0.1–8 ng/mL (buffer)	[47]
Optical Detection (Fluorescence)	S100B	No	Buffer and human serum	40 µL	~30 min	0.01 µg/mL (buffer)	0.03–1 µg/mL (buffer)	[76]
Optical Detection (Fluorescence)	GFAP IL-6 IL-8	Yes	Buffer, human serum, and human whole blood	100 µL	~40 min	GFAP (serum): 125 pg/mL IL-6 (buffer): 437 pg/mL IL-8 (buffer): 2 pg/mL	GFAP (serum and whole blood): 100–10,000 pg/mL IL-6 (buffer): 1000–25,000 pg/mL IL-8 (buffer): 10–1000 pg/mL	[46]

Notes: ^a^ LLODs were listed as ranges and were not separated by measurement technique, sample matrix, or biomarker. ^b^ LLODs and ranges were taken as the cutoff concentrations for each logic condition assignment. ^c^ Sample volume was determined based on the flow rate (1 µL/min) and analysis time. ^d^ Measurements were run in parallel using separate electrodes on the same chip. ^e^ Three separate SERS substrates (one for each biomarker) were incorporated into the microfluidic chip. ^f^ Stated that 1–2 drops of capillary whole blood were used in the microfluidic chip. ^g^ This was the only concentration of GFAP that was tested. ^h^ A range of values was not provided for CRP. ^i^ This range of values is an estimate as it was pulled from a graph.

**Table 3 biosensors-11-00319-t003:** Summary of late-stage sensors for traumatic brain injury protein biomarker measurement.

Device	Detection Technique	Biomarker(s)	Multiplex	Sample Type	Sample Volume	Analysis Time	Lower Limit of Quantitation (LLOQ)	Range	Clinical Studies
Banyan BTI^TM^	Optical Detection (Chemiluminescence)	GFAP UCH-L1	No	Human serum [93]	250 µL [93]	>2 h [93]	GFAP: 10 pg/mL UCH-L1: 80 pg/mL [93]	GFAP: 10–320 pg/mL UCH-L1: 80–2560 pg/mL [93]	[94,95,96,97,98,99,100,101,102,103,104,105]
Abbott i-STAT Alinity	Electrochemical Detection (Amperometric)	GFAP UCH-L1	Yes	Human plasma [106]	20 µL [106]	15 min [106]	GFAP: 23 pg/mL UCH-L1: 70 pg/mL [106]	GFAP: 30–10,000 pg/mL UCH-L1: 200–3200 pg/mL [106]	[17,23]
Quanterix Simoa^®^	Optical Detection (Fluorescence)	GFAP UCH-L1 Tau NF-L NSE [107]	Yes	Human CSF, serum, and plasma [108]	100–152 µL [108,109]	2 h and 30 min per 96-well plate [110]	GFAP: 0.467 pg/mL UCH-L1: 5.45 pg/mL Tau: 0.053 pg/mL NF-L: 0.241 pg/mL NSE: 9.88 pg/mL [108,109]	GFAP: 0–4000 pg/mL UCH-L1: 0–40 ng/mL Tau: 0–400 pg/mL NF-L: 0–2000 pg/mL NSE: 0–120 ng/mL [108,109]	[111,112,113,114]

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
