# Peer review of "The Current State of Traumatic Brain Injury Biomarker Measurement Methods"

_biosensors, 2021, doi:10.3390/bios11090319_

Round 1

Reviewer 1 Report

For the reader interested in this topic, the description of the different methods and devices for the quantification of cerebral biomarkers carried out by the authors is very complete and documented. Likewise, the review on the possibilities of applicability of biomarkers in clinical practice is of interest, especially in situations where their use could be of greater interest, such as the scene of the accident. I think it gathers interest and quality to be published.

The review is very complete, and meets the objectives set by the authors in the introduction. The article can be shortened avoiding redundancies in the text, with what it would gain in its reading. As an example, it is repeated up to four times (lines 78-80, 98-99, 488 and 507 ) “As of June 2021, 78 only  GFAP  and  ubiquitin  c-terminal  hydrolase  L1  (UCH-L1)  are  FDA  cleared  “. Likewise, since the ideal characteristics that the molecules and apparatus should meet are described in line 107 and following, it is not necessary to repeat this in the discussion, as it happens in line 497.

Reviewer 2 Report

Authors present a well constructed review on sensors in development for biomarker indications for TBI- and specifically mild TBI.

Minor points I would like to see addressed:

Authors state (line 86-88 of intro) that they will not review devices/mechanisms that are not specific to TBI indications, I would like to see the addition of a table of these that were excluded…. Or at least some discussion of some of these devices and what biomarkers identify – why they would not be as sensitive as ‘targeted’ developed sensors for TBI – this would add to the discussion.

Line 126 – why state the analysis time should be <15min … why not <30min, justification of the time frame is warranted – even if it is a suggestion based on clinical management?

A key emphasis in the discussion is around the diagnostic/prognostic power of sensors, and how they would be utilised to inform clinical management of whether to send a patient for a head CT or not. Line 411 authors state that 2h ‘physicians cannot wait that long to order a head CT’ – is this based on specific clinically published guidelines on management of TBI. I think to state that clinicians ‘cannot wait’ is overstating- especially in mTBI scenarios.

Line 431 – authors mentioned potential deployment in the USA “Fall 2021” – be mindful that this is vague for readers outside of the USA/ in southern hemisphere- please change to a specific month(s).

Line 511-514 – authors could concede in the discussion that while device development needs to align with current FDA approved biomarkers- scientific innovation will be drawn towards novel biomarkers- as such development should/will include multiplexing systems.

Pg 516 onwards… authors mention about selection of biofluid, but don’t go on to discuss the potential errors involved in whole blood analysis. **potential for haemolysis from RBCs and therefore analyte interference.

I found figure 6 and figure 7 redundant – they offer no additional value to the review above what is presented in the text.

The discussion focuses on biomarker detection/applicability to identifying which TBI patients may need a head CT and who does not need head CT. The authors do not then expand on the relevance if this- why is it pertinent to distinguish between these patients? Is it a simple costing to healthcare systems? Is it that head CT scans require contrast dyes that have the potential to be harmful of pateints? What is the key benefit to distinguish the mTBI patients that will require scans?
